# Methyl Donors, Epigenetic Alterations, and Brain Health: Understanding the Connection

**DOI:** 10.3390/ijms24032346

**Published:** 2023-01-25

**Authors:** Rola A. Bekdash

**Affiliations:** Department of Biological Sciences, Rutgers University, Newark, NJ 07102, USA; rbekdash@newark.rutgers.edu

**Keywords:** brain, epigenetics, methyl donors, SAM, stress

## Abstract

Methyl donors such as choline, betaine, folic acid, methionine, and vitamins B6 and B12 are critical players in the one-carbon metabolism and have neuroprotective functions. The one-carbon metabolism comprises a series of interconnected chemical pathways that are important for normal cellular functions. Among these pathways are those of the methionine and folate cycles, which contribute to the formation of S-adenosylmethionine (SAM). SAM is the universal methyl donor of methylation reactions such as histone and DNA methylation, two epigenetic mechanisms that regulate gene expression and play roles in human health and disease. Epigenetic mechanisms have been considered a bridge between the effects of environmental factors, such as nutrition, and phenotype. Studies in human and animal models have indicated the importance of the optimal levels of methyl donors on brain health and behavior across the lifespan. Imbalances in the levels of these micronutrients during critical periods of brain development have been linked to epigenetic alterations in the expression of genes that regulate normal brain function. We present studies that support the link between imbalances in the levels of methyl donors, epigenetic alterations, and stress-related disorders. Appropriate levels of these micronutrients should then be monitored at all stages of development for a healthier brain.

## 1. Introduction

Environmental factors such as diet and stressors have substantial effects on brain health [1]. The impact of these factors could be long-lasting if we are exposed to them during early life. Methyl-donor micronutrients play an important role in normal brain development and function [2,3,4,5]. Micronutrients such as choline, betaine, folic acid, methionine, and vitamins B6 and B12 have been shown to modulate the epigenome [5]. They are critical players in the one-carbon metabolism which consists of chemical reactions and some of these reactions lead to the formation of S-adenosylmethionine (SAM). SAM is a universal methyl donor for key epigenetic mechanisms such as DNA methylation and histone methylation [6]. These mechanisms regulate gene expression and function without altering the gene sequence and play a key role in human health and disease [7].

Studies have shown that methyl-donor micronutrients can act as neuroprotectants in the developing brain by causing epigenetic alterations in key neuronal genes [5,8,9]. For example, changes in DNA methylation and changes in histone marks have been reported in stress-related disorders [10,11,12,13]. Exposure to stressors during early development has been linked to epigenetic changes in different brain regions that play a role in cognitive function or regulation of the stress response and behavior [14,15,16]. Early life stressors could cause epigenetic programming of stress-related genes with long-term effects on the functionality of the stress or the hypothalamic–pituitary–adrenal (HPA) axis and other neuronal networks [17,18]. Research studies conducted in human and animal models show a link between early supplementation of dietary components with methyl donors and changes in cognitive functions and behavior [9,19,20,21,22].

In this review, we will focus on the role of methyl-donor micronutrients in the one-carbon metabolism and how they impact gene expression regulation by epigenetic mechanisms. We will summarize those studies that show the effects of early life stressors on brain function and report whether the supplementation of these micronutrients can cause epigenetic alterations of key genes related to cognitive functions and behavior and hence influence disease progression or prevention.

## 2. Epigenetic Mechanisms

Epigenetic mechanisms regulate gene expression and function by altering the chromatin structure without altering the base sequence of DNA. These mechanisms are interrelated and include DNA methylation, posttranslational modifications of histones or histone modifications, chromatin remodeling, and the role of microRNAs [23,24,25,26,27] (Figure 1). These mechanisms can be induced by environmental factors, could be reversible, and have been linked to the etiology of many diseases or disorders such as cancers, cardiovascular diseases, metabolic disorders, and neurological disorders [7]. It is, therefore, critical to understand the role of nutrients, as environmental factors, in altering gene expression by their epigenetic mechanisms and how these alterations that happen at critical stages of development are linked to human health and disease.

DNA methylation plays essential roles in a range of biological functions in mammals such as the silencing of transposable elements, regulation of gene expression, DNA replication, parental imprinting, X-chromosome inactivation, control of cellular differentiation, normal embryonic development, normal brain development, and brain plasticity [28,29]. DNA methylation is a covalent inert modification that chemically modifies the DNA without altering its charge. It does that by adding the methyl group CH3 to the carbon 5 on cytosine which is located next to guanine in the CpG dinucleotides, which leads to the formation of 5-methylcytosine (5-mC) as a nitrogenous base [25,30,31]. This covalent modification is catalyzed by the activity of DNA methyltransferases (Dnmt1, Dnmt3a, Dnmt3b, and Dnmt3L) that utilize S-adenosylmethionine (SAM) as a universal methyl-donor. Structurally, these mammalian enzymes share a conserved C-terminal catalytic domain important for their enzymatic activity except for Dnmt3L. They also contain an N-terminal regulatory domain except for Dnmt2. The N-terminal domain is essential for protein–protein interactions such as interactions of Dnmts with proteins or effectors that are involved in the modulation of chromatin structure and function [32]. These enzymes are expressed in a spatial and temporal manner and have been shown to be abundantly expressed in the brain. Although DNA methylation is considered one of the most stable and inert epigenetic marks in the mammalian genome, it can be reversible [33,34]. This was demonstrated with the discovery of DNA demethylases such as the ten-eleven-translocation (TET) family of proteins. The activity of these enzymes led to the discovery of the 5-hydroxymethylcytosine (5-hmC) as a modified nitrogenous base with a role in cellular differentiation, aging, and cancer [35,36,37]. The majority of CpGs are concentrated in the gene promoter and in the first exon of a gene they are known as CpG islands and are often unmethylated. The abnormal methylation of these CpGs is linked to gene repression [30]. Studies have shown that the supplementation of specific micronutrients that act as methyl donors can cause global or gene-specific changes in methylation in the developing and mature brain, thus altering gene expression and susceptibility to diseases [3,5,38].

Histone modification is another epigenetic mechanism that chemically modifies the N-terminal tail of histones. In this way, it alters the interaction of DNA with histone proteins, around which the DNA is wrapping itself resulting in changes in gene expression. The switch between chromatin compaction and chromatin relaxation is regulated by the ability of the N-terminal tail to perform malleable posttranslational modifications (PTMs) at specific amino acid residues along this tail. These modifications are not random but occur at specific residues such as lysine (K), arginine (R), serine (S), or threonine (T) and accordingly, can alter the accessibility of transcription factors to their binding sites [39,40]. The best-understood and well-characterized histone modifications include methylation, acetylation, phosphorylation, ubiquitination, and sumoylation. These modifications are written by “writers”, erased by “erasers”, and read by effector proteins. Most histone methylation and acetylation happen at the lysine and arginine residues of histone H3 and H4; lysine methylation is catalyzed by histone methyltransferases (HMTs/KMTs) and lysine demethylation by histone demethylases (HDMs/KDMs). Similarly, histone acetylation is regulated by histone acetyltransferases (HATs/KATs) and deacetylation by histone deacetylases (HDACs) [40,41,42,43,44]. The epigenetic factors of microRNAs represent another layer of gene expression regulation. They are described as short RNAs that do not code for proteins but regulate the expression of many protein-coding genes. They require the RNA-induced silencing complex (RISC) to guide them to their target which is the 3′-untranslated region (3′UTR) of a gene. Once they reach their target, they cause gene silencing by either degrading the mRNA or halting its translation [26,45,46]. These epigenetic factors are abundantly expressed in the developing and mature brain and can modulate gene expression at different stages of neuronal development in diverse organisms [47,48]. The dysregulation in microRNA expression has been associated with several diseases or disorders [49], including neurological disorders [48,50,51,52]. These epigenetic mechanisms described above are now considered plausible mechanisms in the etiology of many diseases that are environmentally induced and not necessarily caused by genetic factors.

## 3. One-Carbon Metabolism

One-carbon metabolism consists of chemical reactions that are catalyzed by several enzymes with the contribution or presence of methyl-donor micronutrients. These reactions support several pathways such as nucleotide metabolism, redox state, neurotransmitters synthesis such as acetylcholine, and regulation of epigenetic mechanisms via the formation of SAM [53,54,55,56]. Folate and methionine cycles are two main components of the one-carbon metabolism [54], among which the methyl-donor micronutrients such as folate, methionine, choline, betaine, and vitamins B6 and B12 are critical players. There is a link between the changes in the levels of these micronutrients and epigenetic alterations (Figure 2). Table 1 provides a list of methyl-donor micronutrients that contribute to the one-carbon metabolism.

The main outcome of the methionine cycle is the formation of SAM that powers methylation reactions such as DNA methylation and histone methylation. SAM donates the methyl group to the epigenetic machinery such as DNA methyltransferases (DNMTs) that methylate the DNA or to histone methyltransferases (HMTS/KMTs) that methylate histones [6,38,53] (Figure 2). After the transfer of the methyl group to the substrate, SAM is converted to S-adenosylhomocysteine (SAH). Under normal conditions, SAH is hydrolyzed to yield adenine and homocysteine by S-adenosylhomocysteine hydrolase (SAHH) (Figure 2). On the other hand, the elevation in SAH levels compared to SAM has an inhibitory effect on DNA methyltransferases with the potential to alter DNA methylation [74,75]. Via the catalytic activity of cystathionine β-synthase (CBS), homocysteine with serine can form cystathionine, which is further catalyzed to form products with antioxidant properties. Methionine can be regenerated by the activity of two enzymes, betaine-homocysteine methyltransferase (BHMT) and methionine synthase (MS). Betaine-homocysteine methyltransferase can transfer a methyl group from betaine, as a choline precursor or derived from the diet, and generate methionine and dimethylglycine (DMG). The folate metabolism leads to the formation of purine and 5-methyltetrahydrofolate (5-MTHF) using 5-methyltetrahydrofolate reductase (5-MTHFR). It is worth noting that methyltetrahydrofolate reductase (MTHFR) gene polymorphism alters its enzymatic activity and has been linked to altered cognitive functions [76]. Another enzyme, 5-methyltetrahydrofolate-homocysteine methyltransferase or methionine synthase (MS), a VitB12-dependent enzyme, can transfer a methyl group from 5-methyltetrahydrofolate (5-MTHF) to homocysteine and in this manner produces tetrahydrofolate (THF) and methionine (Figure 2). Methionine is adenylated by methionine adenosyltransferase (MAT) to generate SAM [53]. Dietary methyl donors such as choline, betaine, folic acid, methionine, and the B vitamins are interconnected in this one-carbon metabolism. Changes in their levels can alter gene expression and regulation by altering the levels of SAM, linking changes in the intake of dietary methyl donors to alterations in cellular functions.

## 4. Dietary Methyl Donors, Epigenetic Alterations, and Stress-Related Disorders

Studies have shown that early life experiences cause a developmental programming of the hypothalamic–pituitary–adrenal (HPA) axis or stress axis and behavioral responses to stressors [17,77,78]. This developmental programming of the stress axis induced by early life experiences could be explained by changes in the expression of stress-related genes by epigenetic mechanisms such as DNA methylation with long-term neurobehavioral outcomes [79,80,81,82]. The supplementation or deficiency in the levels of methyl-donor micronutrients during early life can have an impact on offspring brain development with long-term effects on behavior [83,84,85,86]. In this section, we will describe the programming of the stress axis through early life experiences and the role of methylation, then we will summarize research studies that explain this intricate link between micronutrients, changes in methylation, and stress-related or neurodevelopmental disorders.

### 4.1. HPA Axis Programming by Early Life Stress and the Role of Methylation

The hypothalamic–pituitary–adrenal (HPA) axis represents the organism’s neuroendocrine response to stressors. Upon HPA axis activation, the immune system and the nervous system are also activated. This results in the release of stress mediators such as endocrine hormones, cytokines, and neurotransmitters. These mediators enable the organism to react and respond to stress leading to an adaptive or maladaptive response [87,88,89]. The main mediator that is released upon HPA axis activation is glucocorticoid GC (or cortisol), which exerts a negative feedback mechanism at the levels of the anterior pituitary, hypothalamus, and hippocampus. GC can cross the blood–brain barrier and bind to its glucocorticoid receptor (GR), with higher affinity to the mineralocorticoid receptor (MR). Glucocorticoid receptors and mineralocorticoid receptors are widely expressed in limbic regions such as the prefrontal cortex, hippocampus, amygdala, and hypothalamus (Figure 3). The binding of GC to its receptor induces signaling mechanisms that regulate the stress response and behavior via the activation or repression of stress-related genes [90]. Aberrant release of GC or a blunted HPA axis response is harmful and has been shown to contribute to psychopathology [89,90,91].

The dysregulation of the HPA axis and imbalances in the levels of GCs and the expression of its receptors in limbic regions have been linked to psychiatric disorders [92,93,94]. In particular, exposure to stress during early life is considered one of the main early life experiences that can result in long-term consequences such as neurobehavioral changes that may develop into major psychosis [95,96,97,98]. One plausible explanation that mediates this link between early life experiences and long-term neurobehavioral outcomes are epigenetic mechanisms that chemically modify the expression of stress-related genes via methylation [1,12,81,99,100]. Genome-wide changes in methylation, hypo- and hypermethylated sites, were reported in the brain tissues of suicidal individuals or individuals with a history of early life adversity [101,102]. 

The brain is plastic, especially early in development where exposure to adaptive or maladaptive environmental factors can have positive or negative long-term effects on health that could pass to the next generations [103,104]. We evaluate here evidence of the role of DNA methylation in the embedding of early life experiences and the role of nutrition. One of the most studied brain regions in humans and rodents that demonstrates the long-term effects of early life experiences on behavior is the hippocampus. For example, the effects of maternal care in the form of licking and grooming during postnatal life impact the behavior and the stress response of offspring during adulthood. The alterations in the offspring stress response later in life were associated with a change in the methylation status of the CpG islands of the glucocorticoid receptor (GR) (*NR3C1*) gene promoter that altered the binding of the nerve growth factor-induced protein A (NGF1-A) transcription factor to its binding site in the GR promoter and altered GR gene expression in the rat hippocampus. This study demonstrated the epigenetic effects of maternal care on the offspring’s stress response in relation to changes in the methylation of the GR gene promoter [105]. Similarly, human postmortem hippocampal tissues of suicidal victims with a history of childhood abuse or maltreatment showed an increase in the methylation of *GR* with a decrease in its expression [106]. The effects of early life experiences impact other stress-related genes besides GR. Exposing infant rats during early postnatal life to stress altered the methylation status and the expression of the brain-derived neurotrophic factor *Bdnf* gene in the adult prefrontal cortex. This alteration persisted in the next generation of infant rats demonstrating the transgenerational transmission of the effect of early stress on brain-derived neurotrophic factor *Bdnf* and behavior [107]. Another study conducted in mice showed the effects of early life stress (ELS) on neuroendocrine function in the hypothalamic paraventricular nucleus (PVN). Early life stress resulted in hypomethylation of the promoter of the *Vasopressin gene* (*Avp*) with an increase in its expression. These epigenetic changes were associated with an elevation in blood corticosterone levels, an impaired ability of offspring to cope with stress, and an impaired memory [108]. An interesting study conducted in Wistar Kyoto rats, a highly stress-susceptible strain with anxiety/depression-like phenotypes and a hyperactive HPA axis, showed a positive effect of ELS on offspring neurodevelopment during adulthood. Maternal separation for 180 min from postnatal days 1 to 14 caused global hypermethylation in the rat hippocampus but not in other limbic regions with reduced methylation in the insulin receptor and its downstream targets, known to have a role in memory and neuronal survival. Several other genes that are linked to cell proliferation, tyrosine kinase signaling, axonal guidance, synaptogenesis, and transmission were differentially methylated. Interestingly, the GR (*Nr3c1*) gene methylation was not altered in this study. These methylation changes were associated with diminished depressive or anxiety-like behavior, increased exploratory behavior, and increased sociability in offspring suggesting stress resilience in adulthood as an adaptive response of offspring to persistent ELS. Dietary methyl-donor supplementation for four weeks during adulthood had anxiolytic and antidepressant effects in these rats with improved cardiovascular responses to stress [109]. 

Methyl-CpG-binding protein (MeCP2) has been implicated in the etiology of the developmental disorder, Rett syndrome (RTT). Rett syndrome mouse models show altered corticosterone response to stress, dysregulated levels of the stress hormone corticotropin-releasing hormone or factor (Crh/Crf), and dysregulation of the stress axis [110,111,112]. In this context, one study investigated the effects of ELS such as maternal separation (MS) from postnatal days 3 to 21 in MeCP2 heterozygote female mice (MeCP2-het-MS) and wild-type (WT-MS) mice on anxious behavior during adolescence using behavioral tests such as open field, the forced swim test, and elevated plus maze. At six weeks of age, MeCP2-het-MS mice show less anxiety and less depressive-like behaviors compared to WT-MS mice, with reduced neuronal activation in the PVN, as depicted by the immunoreactivity of c-fos/Avp and c-fos/Crh. These findings indicate the role of MeCP2 functionality on stress axis regulation and its impact on emotional behavior and neuronal activity later in life [113].

Interestingly, one study reported the hypermethylation of a distal cytosine guanine island (CGI) shore of the GR (*Nr3c1*) in Crh-producing neurons in the PVN of the hypothalamus, a brain region that is involved in stress regulation, leading to upregulation of *GR* and thus preventing the elevation of Crh in response to stress in adulthood [114]. Other stress-related genes that are found to be altered epigenetically in response to early life stress are *vasopressin* (*Avp*) and *Crh/Crf* in the hypothalamic PVN. For example, maternal separation in mice resulted in the hypomethylation of CpG sites along the enhancer of *Avp* leading to an increase in its expression with an increase in stress responsiveness due to an elevation in corticosterone. This elevation led to altered feedback inhibition of the HPA axis response leading to a hyperactive stress response [108]. Interestingly, this hypomethylation of *Avp* was linked to the phosphorylation of MeCP2 and an altered ability of phosphorylated MeCP2 to bind to the *Avp* enhancer and recruit DNA methyltransferases to cause gene repression [115]. Early prenatal stress caused hypomethylation of CpG sites of the *Crf* promoter in the hypothalamus and central amygdala of mice with elevated levels of corticotropin-releasing factor (CRF), suggesting dysregulation of their stress axis during adulthood [116].

What about Bdnf in the context of ELS and epigenetic alterations? Epigenetic regulation of *Bdnf* expression is affected in response to ELS. For example, the release of MeCP2 repressor complex from the *Bdnf* promoter results in its demethylation with an increase in *Bdnf* expression [117], and the epigenetic mediator microRNA-132 has been shown to downregulate *MeCP2* expression and indirectly reduce Bdnf levels in the hippocampus of a rat model of chronic stress-induced depression [118]. Epigenetic regulation of *Bdnf* has been linked to neuroplasticity and neuronal activity in mature hippocampal neurons [119]. Studies have demonstrated that *Bdnf* expression is critical for normal dendritic branching in the hippocampus and amygdala implicating the role of Bdnf in major behavioral correlates of stress disorders such as anxiety and depressive-like behaviors [120,121]. The type, intensity, and duration of the stressor, the developmental period studied, the brain region involved, and the rodent strain used in studies may have resulted in different interpretations and results in the scientific field. For example, postnatal stress resulted in *Bdnf* hypermethylation and reduced *Bdnf* expression in the rat adult prefrontal cortex that persisted in the next generation [107], whereas prenatal stress resulted in decreased *Bdnf* expression and hypermethylation at *Bdnf* exon IV in the rat amygdala and hippocampal regions during adulthood with an increased expression of *Dnmt1* and *Dnmt3a* [122].

### 4.2. Potential Neuroprotective Effects of Dietary Methyl Donors: More or Less?

The etiology of many diseases is now believed to be caused by not only genetic factors but also environmental factors via epigenetic mechanisms that do not follow mendelian inheritance patterns [24,123]. Dietary methyl-donor micronutrients that contribute to the one-carbon metabolism have been shown to be critical during development as they contribute to the formation of SAM which is involved in methylation reactions that are essential for brain health across the lifespan [5]. In this section, we will present evidence from findings that demonstrate the effects of the optimal levels of these micronutrients during specific stages of development on human health and disease. 

Folate maternal intake is quite important to decrease the incidence of neural tube defects (NTDs) in children and its deficiency has been linked to many diseases including anemia, atherosclerosis, psychiatric disorders, and cancer [93,124]. Consistent results in clinical trials showed the beneficial effects of using folate in conjunction with other pharmacological interventions in mitigating the effects of psychiatric disorders such as schizophrenia, bipolar disease, autism, and attention-deficit hyperactivity disorder [125]. Several human studies demonstrated the correlation between maternal inadequate intake of micronutrients and altered neurodevelopment such as brain defects, altered behavior, altered cognition, and potential contribution to psychiatric disorders [10,126,127]. A large study conducted in humans showed that maternal intake of folate and VitB12 during the first trimester of pregnancy correlated with a higher score in cognitive measures in children at age 3 years [128]; in addition, a Norwegian study demonstrated a reduced risk of language delay in children at 3 years of age with folate maternal intake during early pregnancy [129]. The neuroprotective effects of micronutrients are not only evident during early life. Several studies linked the intake of specific micronutrients to cognitive performance during adolescence. For example, higher dietary folate intake positively correlated with academic performance in adolescents [127,128,129,130]. Animal studies showed comparable findings. For example, prenatal folate deficiency during the gestational days GD11-GD17 in rats is linked to structural changes in the brain such as a reduction in the number of progenitor cells in the fetal neocortex [131]. VitB deficiency correlated with the elevation of homocysteine in neurons and astrocytes in specific brain regions such as the striatum, hippocampus, and cerebellum. Homocysteine-positive cells showed markers of death. These rats had altered motor function and altered cognitive functions such as learning and memory deficits during adulthood [132]. 

In an early life induced model of depression in rats, maternal separation for 180 min from postnatal days PD2 to PD21 altered the levels of total high-density lipoprotein-cholesterol (HDL-cholesterol) levels and increased depressive-like behaviors in offspring, as measured by an increased immobility time in the forced swimming test displayed by these rats. An eighteen-week supplementation of choline, betaine, folate, and VitB12 at PD60 reduced the depressive-like behavior in offspring and increased total DNA methylation in the hippocampus, as measured by the levels of 5-methylcytosine [21]. The anti-depressive action of methyl donors was demonstrated in chronically high-fructose-treated rats, an animal model of anxiety and mood disorders. Eight weeks of methyl-donor supplementation at 4 weeks of age in female rats reduced oxidative stress, as measured by the nitrite content in the rat hippocampus, reduced anxiety-like behavior in the elevated plus maze test, and depressive-like behavior in the forced swimming test, reinforcing the notion that methyl donors could act as nutri-therapeutic agents in mitigating the effects of stress-related disorders such as anxiety and depression [133]. 

The neuroprotective effects of methyl donors of the one-carbon metabolism were also reported in humans with and in animal models of neurodegenerative disorders such as Alzheimer’s disease (AD). AD is one of the most common types of cognitive impairment with aging and is associated with altered one-carbon metabolism. Its neuropathology is characterized by the accumulation of amyloid beta peptide plaques and the hyperphosphorylation of the microtubule-associated protein tau, causing the formation of neurofibrillary tangles [134]. Recent studies support the notion that diets rich in micronutrients that play a role in the one-carbon metabolism have promising effects in mitigating AD pathogenesis [67,135,136,137,138,139]. In the context of methyl donors, an increase in plasma homocysteine, and hence SAH as a methyltransferase inhibitor, was reported in AD brain samples and correlated with cognitive impairment [140]. The association between VitB and AD is still debatable. Presenilin (PSEN1) plays a role in increasing secretase activity that cleaves the amyloid precursor protein (APP) into amyloid beta peptides. Studies have linked VitB deficiency to hypomethylation of the *PSEN1* promoter and an increase in its expression in an AD TgCRND8 mouse model. Supplementation of SAM to these mice reversed the effects on *PSEN1* hypomethylation and expression, tau phosphorylation, and reduced amyloid beta peptide production [141,142]. On the other hand, folic acid supplementation in an amyloid precursor protein/presenilin (APP/PSNE1) transgenic mouse model of AD reduced the levels of amyloid beta proteins as this correlated with increased activity of Dnmt1 and the hypermethylation of *PSEN1* and *APP* promoters which correlated with a decrease in their expression [143]. Deficiency or disruption in glutathione (GSH) levels, an antioxidant that is produced by homocysteine in the transsulfuration pathway (Figure 2), has been linked to cognitive decline in AD patients and contributes to AD-related oxidative stress [144,145]. In the context of betaine, another methyl-donor, human studies showed that betaine supplementation in AD patients results in: decreased levels of homocysteine, phosphorylated tau, and amyloid beta accumulation; a reduction in blood inflammatory markers such as interleukin-1 beta (IL-1β) and tumor necrotic factor-alpha (TNF-α); an increase in the levels of memory-related proteins such as NR1, NR2A, and NR2B (NMDA receptors); an increase in the levels of synaptic proteins such as synaptophysin, synaptotagmin, and phosphorylated synapsin I [146]. 

What about choline? Choline is the main precursor for betaine and serves as a methyl donor for the formation of SAM via the methionine cycle. Choline is neuroprotective, can be derived from food, has essential functions related to the production of neurotransmitters such as acetylcholine that regulates cholinergic signaling, and maintains the integrity of cellular membranes via the formation of phosphatidylcholine [147] (Figure 2). Dietary supplementation of choline in the amyloid precursor protein/presenilin1 (APP/PS1) mouse model of AD improved spatial memory in the Morris water maze and reduced the processing of APP to amyloid beta peptides with a reduction in microglia activity, which is known to play a role in neuroinflammation, seen in AD, suggesting the potential benefits of diets rich with choline on brain function [148]. Since acetylcholine is quite essential for cholinergic signaling in the brain, acetylcholine esterase inhibitor, the enzyme that degrades acetylcholine, is currently used as a drug for AD treatment, although its use showed side effects and did not prevent AD progression due to low efficacy [149]. Choline perinatal supplementation (fetal and early postnatal) in AD mouse models showed positive improvements at several levels. These mice showed in the hippocampus: reduced levels of amyloid beta proteins; reduction in amyloid plaque accumulation; elevation of choline acetyltransferase (ChAT), the enzyme that forms acetylcholine from choline; a decrease in the levels of glial fibrillary acidic protein (GFAP) proteins, a marker of gliosis, suggesting the importance of choline intake during pregnancy and shortly after birth [150]. Another study demonstrated the beneficial effects of choline supplementation in an AD mouse model (APP/PS1 mice) during adulthood. Supplementing these mice from 2 to 11 months of age with choline resulted in reduced anxiety and improved spatial and learning memory in several behavioral tests. This correlated with reducing amyloid beta accumulation in the cortex and the hippocampus with a decrease in the levels of microglia activation markers, restoration of choline and acetylcholine levels in the cerebral cortex and the hippocampus, and increased ChAT-positive cholinergic neurons in the basal forebrain and amygdala. Moreover, choline effects on synapses were also determined. Choline supplementation increased the levels of synaptophysin and the postsynaptic density protein (PSD95) in the hippocampus of this AD mouse model [151].

## 5. Conclusions

Early life stress can have adverse long-term effects on health with an increased possibility of developing neuropsychiatric disorders later in life. The ability to respond and react to stress involves the regulation of the HPA axis and the involvement of brain regions and neuronal networks that belong to the limbic system. Our mental health such as our adaptation to stress and our resilience in the face of adversity is dependent on the time of exposure to stressors during development, the duration, intensity, and the type of stressor. It also includes the genetic make-up of an individual and other environmental factors. These factors greatly impact brain health across the lifespan. Epigenetic mechanisms are considered plausible mechanisms that explain this intricate relationship between our genes and our environment that may impact behavior and brain health. Studies have shown that supplementation of methyl donors during early life is essential for normal brain growth and development and has long-lasting effects on mental health. The regulated consumption of these nutrients later in life has been shown to improve cognitive functions and mitigate the symptoms of neurodegenerative disorders when used with other pharmaceutical interventions. The micronutrients of methyl donors in the one-carbon metabolism have been reported to be neuroprotective, can cause global or gene-specific changes by epigenetic mechanisms, and could be used in a regulated manner to mitigate the symptoms of neurodegenerative diseases and stress-related disorders at early stages of prognosis. The dosage and the timing of intake of these micronutrients in healthy individuals should be monitored by a health professional to prevent unwanted side effects or overdosage, as these micronutrients play crucial roles in many physiological processes and may have an impact on brain health.

## Figures and Tables

**Figure 1 ijms-24-02346-f001:**
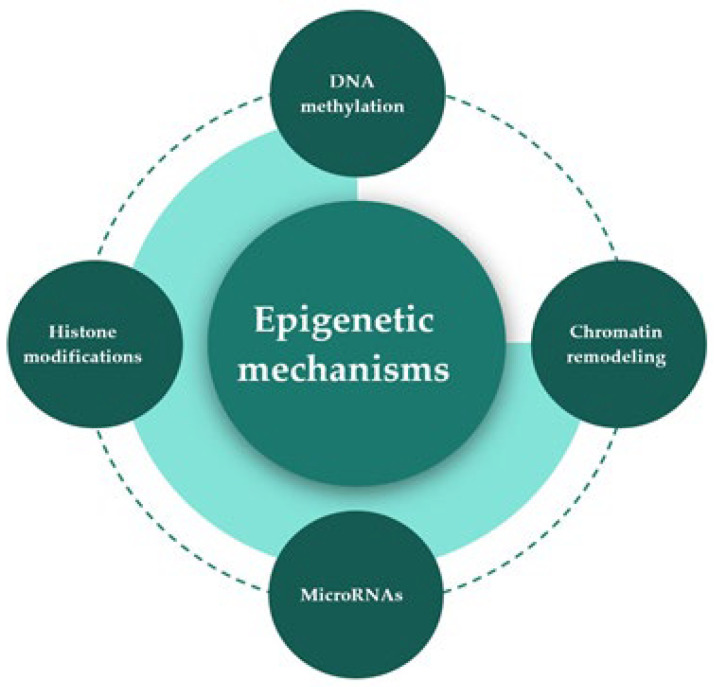
Epigenetics mechanisms. This is a schematic view of the epigenetic mechanisms that regulate gene expression. They are interrelated and include DNA methylation, histone modifications, chromatin remodeling, and the role of microRNAs.

**Figure 2 ijms-24-02346-f002:**
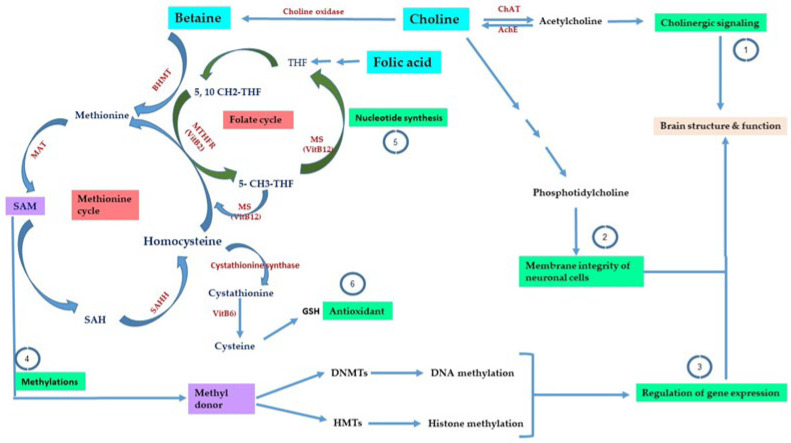
One-carbon metabolism. This figure shows the main components of the one-carbon metabolism, the folate cycle, the methionine cycle, and the contribution of micronutrients such as choline, betaine, folate, methionine, VitB6, and VitB12 in several biological processes. Enzymes and vitamins are written in red and biological processes are highlighted in green. Micronutrients are highlighted in blue. THF: tetrahydrofolate, SAM: S-adenosylmethionine, SHA: S-adenosylhomocysteine, SAHH: SAH hydrolase, BHMT: betaine-homocysteine methyltransferase, MAT: methionine acetyltransferase, DNMTs: DNA methyltransferases, HMTs: histone methyltransferases, MS: methionine synthase, MTHFR: methyltetrahydrofolate reductase, ChAT: choline acetyltransferase, AchE: acetylcholine esterase, GSH: glutathione. Adapted from Ref. [5].

**Figure 3 ijms-24-02346-f003:**
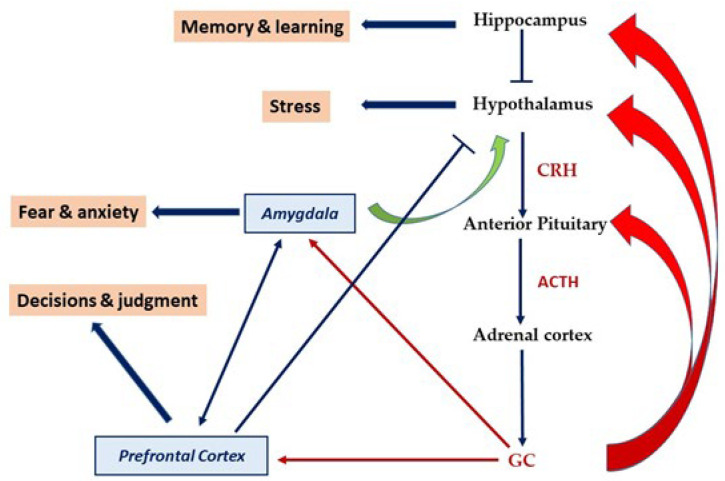
HPA axis. This figure represents the HPA axis or the stress axis where glucocorticoids (GCs) are the outcome of its activation. GC exerts feedback at the levels of the anterior pituitary, hypothalamus, and hippocampus to regulate the stress response. GC receptors are expressed in the hippocampus, hypothalamus, prefrontal cortex, and amygdala, which are considered part of the limbic system. CRF/CRH: corticotropin-releasing hormone or factor, ACTH: adrenocorticotropin hormone, GC: glucocorticoid. Red arrows on the right side mean negative feedback. Green arrow means positive feedback.

**Table 1 ijms-24-02346-t001:** This table summarizes the functions of methyl donors that contribute to the one-carbon metabolism.

Methyl Donors	Function	References
Methionine	Precursor for SAM formation, maintenance of the redox state, and brain health.	[57]
Choline	Regulation of cholinergic signaling, maintaining cellular membrane integrity, and contributing to the formation of SAM.	[58,59,60,61]
Betaine	Choline precursor, a methyl-donor in the BHMT pathway, and anti-inflammatory functions.	[62,63,64]
Folic acid	Normal brain development, nucleotide synthesis, and prevention of neural tube defects.	[65,66,67]
Vitamin B12	Nucleotide synthesis, antioxidant properties, and maintaining brain health.	[68,69,70]
Vitamin B6	Maintenance of the redox state and brain health. Role in transamination and decarboxylation reactions required for the metabolism of several neurotransmitters. Nucleotide synthesis and protein/lipid metabolism.	[71,72,73]

## Data Availability

Not applicable.

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
