# Peer review of "Methyl Donors, Epigenetic Alterations, and Brain Health: Understanding the Connection"

_ijms, 2023, doi:10.3390/ijms24032346_

Round 1

Reviewer 1 Report

As described in Rola A. Bekdash's manuscript titled "Methyl Donors, Epigenetic Alterations, and Brain Health: Understanding the Connection", methyl donors such as choline, betaine, folic acid, methionine, vitamins B6 and B12 have been found to be critical players in one-carbon metabolism and to be neuroprotective as well. For normal cellular processes, carbon metabolism consists of a series of interconnected chemical pathways. The methionine and folate cycles contribute to the formation of S-adenosylmethionine (SAM). In epigenetic mechanisms that regulate gene expression and play a role in human health, SAM is the universal methyl donor for methylation reactions such as histone and DNA methylation. Historically, epigenetic mechanisms have been considered a bridge between environmental factors such as nutrition and phenotype. Human and animal studies have revealed the importance of optimal levels of methyl donors in maintaining brain health and behavior throughout life. The presence of imbalances in these micronutrients during critical periods of brain development has been linked to epigenetic alterations in the expression of genes that regulate normal brain function. Several studies have found a link between imbalances in the levels of methyl donors, epigenetic alterations, and stress-related disorders. For a healthy brain, appropriate levels of these micronutrients should be monitored at all stages of development. Regarding the present manuscript, I would like to make a few comments.

-The first issue is the style of the references, please adhere to the MDPI style.

-Section 2 of epigenetics requires more information.

-Why was the focus of the review on carbon metabolism?

-As far as section 4 is concerned, it seems fine to me and the information presented is of importance to the reader.

-There may be a need for more resolution in the figures

-There is a need for more detailed information in the conclusions regarding the health of the brain.

Author Response

Thank you for the reviews of my paper (IJMS- 2149434). The reviews were useful to me in making revisions to the manuscript, and I hope that it will now be suitable for publication.  Please find below my reply below.  Please turn on “Track Changes” and select “All Markup” to see my changes in the revised manuscript.

Reviewer # 1:

The first issue is the style of the references, please adhere to the MDPI style.
I did follow the style that I see in the IJMS doc. Template online.  I would like here the help of the editing journal to fix the reference style if indeed it doesn’t follow the instructions.

  1. -Section2 of epigenetics requires more information. 
    I added more information to that section:  Lines 66-70, lines 76-80, line 85, lines 97-102, line 111, line 113, and lines 115-117.

  2. -Why was the focus of the review on carbon metabolism? The focus of the one carbon metabolism section is to show the contribution of methyl-donors to different physiological processes including DNA methylation and histone methylation, which are two epigenetic mechanisms, the focus of this manuscript.

  3. -As far as section 4 is concerned, it seems fine to me and the information presented is of importance to the reader. Thank you.

  4. -There may be a need for more resolution in the figures. All are high resolution jpg figures, but I changed the colors in figure 3 to make it look clearer.

  5. -There is a need for more detailed information in the conclusions regarding the health of the brain. This information is now added to the conclusion.

Reviewer 2 Report

This study has summarized the role of methyl donors micronutrients on brain heath via the regulation of one-carbon metabolism and how they impact gene expression regulation by epigenetic mechanisms. This is an interesting topic. The paper was well-written and well-presented. The paper could be accepted after following minor revision.

Please add a table to show the chemicals of the Methyl Donors, and their efficiency;

Please add the informaiton of some other nutrients, except Methyl Donors, also could regulation of one-carbon metabolism; 

Line 50, it looks too much space before the 'The...'. Please check the simialr issue throuhout the paper;

Lines 105-108, please add reference for this informations;

What is the meaning of number 1-6 in the Figuer 2?

Please check all the abbreviation was appeared when the full name was firstly used.

Please make explanation for the arrows in the Figure 3.

Author Response

Thank you for the reviews of my paper (IJMS- 2149434). The reviews were useful to me in making revisions to the manuscript, and I hope that it will now be suitable for publication.  Please find below my reply below.  Please turn on “Track Changes” and select “All Markup” to see my changes in the revised manuscript.

Reviewer # 2

This study has summarized the role of methyl donors micronutrients on brain heath via the regulation of one-carbon metabolism and how they impact gene expression regulation by epigenetic mechanisms. This is an interesting topic. The paper was well-written and well-presented. The paper could be accepted after following minor revision.

  1. Please add a table to show the chemicals of the Methyl Donors, and their efficiency;
    Table 1 is now added on line 131.

  2. Please add the information of some other nutrients, except Methyl Donors, also could regulation of one-carbon metabolism; 
    The methyl-donors micronutrients are the ones that contribute to the one carbon metabolism. There are several byproducts of the complex reactions that they regulate as seen in figure 2.

  3. Line 50, it looks too much space before the 'The...'. Please check the similar issue throughout the paper;  
    I reviewed all the manuscript and made sure that no unnecessary space between words is there.

  4. Lines 105-108, please add reference for this informations;
    The citations are now added on line 127.

  5. What is the meaning of number 1-6 in the Figure 2?
    These numbers show the contribution of methyl-donors to several physiological processes such as 1) cholinergic signaling, 2) membrane integrity, 3) regulation of gene expression, 4) methylation, 5) nucleotide synthesis and 6) formation of antioxidants.

  6. Please check all the abbreviation was appeared when the full name was firstly used.
    This is done throughout the manuscript to make sure that the abbreviation appears next to the term when it is first used.  Lines 113, 148, 153, 188, 197, 231, 243, 246265, 267, 277, 291, 337, 345, 375, 385, and 394.

  7. Please make explanation for the arrows in the Figure 3.
    I added an explanation to these arrows in the legends on line 209.

Round 2

Reviewer 1 Report

Thank you for taking into account my previous comments regarding the manuscript titled "Methyl Donors, Epigenetic Alterations, and Brain Health: Understanding the Connection". There is no need for me to make any further comments.